# U-Net and Its Variants Based Automatic Tracking of Radial Artery in Ultrasonic Short-Axis Views: A Pilot Study

**DOI:** 10.3390/diagnostics14212358

**Published:** 2024-10-23

**Authors:** Yuan Tian, Ruiyang Gao, Xinran Shi, Jiaxin Lang, Yang Xue, Chunrong Wang, Yuelun Zhang, Le Shen, Chunhua Yu, Zhuhuang Zhou

**Affiliations:** 1Department of Anesthesiology, Chinese Academy of Medical Sciences and Peking Union Medical College Hospital, Beijing 100730, China; tianyuan95@pumch.cn (Y.T.); langjx09@163.com (J.L.); xueyingi@sina.com (Y.X.); emancipation258@outlook.com (C.W.); shenle@pumch.cn (L.S.); 2Department of Biomedical Engineering, College of Chemistry and Life Science, Beijing University of Technology, Beijing 100124, China; gaoruiyang@emails.bjut.edu.cn (R.G.); shixinran@emails.bjut.edu.cn (X.S.); 3Medical Research Center, Chinese Academy of Medical Sciences and Peking Union Medical College Hospital, Beijing 100730, China; yuelunzhang@outlook.com

**Keywords:** radial artery tracking, perioperative management, deep learning, U-Net

## Abstract

**Background/Objectives:** Radial artery tracking (RAT) in the short-axis view is a pivotal step for ultrasound-guided radial artery catheterization (RAC), which is widely employed in various clinical settings. To eliminate disparities and lay the foundations for automated procedures, a pilot study was conducted to explore the feasibility of U-Net and its variants in automatic RAT. **Methods:** Approved by the institutional ethics committee, patients as potential RAC candidates were enrolled, and the radial arteries were continuously scanned by B-mode ultrasonography. All acquired videos were processed into standardized images, and randomly divided into training, validation, and test sets in an 8:1:1 ratio. Deep learning models, including U-Net and its variants, such as Attention U-Net, UNet++, Res-UNet, TransUNet, and UNeXt, were utilized for automatic RAT. The performance of the deep learning architectures was assessed using loss functions, dice similarity coefficient (DSC), and Jaccard similarity coefficient (JSC). Performance differences were analyzed using the Kruskal–Wallis test. **Results:** The independent datasets comprised 7233 images extracted from 178 videos of 135 patients (53.3% women; mean age: 41.6 years). Consistent convergence of loss functions between the training and validation sets was achieved for all models except Attention U-Net. Res-UNet emerged as the optimal architecture in terms of DSC and JSC (93.14% and 87.93%), indicating a significant improvement compared to U-Net (91.79% vs. 86.19%, *p* < 0.05) and Attention U-Net (91.20% vs. 85.02%, *p* < 0.05). **Conclusions:** This pilot study validates the feasibility of U-Net and its variants in automatic RAT, highlighting the predominant performance of Res-UNet among the evaluated architectures.

## 1. Introduction

Radial artery catheterization is indispensable for a wide range of clinical scenarios, including anesthesiology, intensive care, emergency, and cardiovascular medicine [1,2]. It enables real-time hemodynamic monitoring, facilitates repeated blood tests, and establishes a proactive pathway for life-saving procedures. Ultrasonography has been recommended for radial artery catheterization [3,4], due to its higher success rates, reduced time requirements, fewer attempts needed, and lower complication rates compared to conventional techniques [5,6]. The radial artery tracking (RAT) on the short-axis view is the initial step for ultrasound-guided catheterization using the short-axis approach, the oblique-axis approach [7], the dynamic tip positioning technique [8], and the modified long-axis approach [9]. RAT is also pivotal in evaluating vascular conditions, identifying the appropriate puncture points, and distinguishing challenging cases [10,11].

In addition to radial artery catheterization, RAT may find applications in computational fluid dynamics (CFDs), as image processing is essential in developing engineering tools for hemodynamic prediction, particularly in analyzing vascular structures and blood flow dynamics to assess hemodynamic stability in clinical scenarios [12]. CFD is a well-established method for simulating cardiovascular hemodynamics, provides insights into parameters like wall shear stress, pressure distribution, and flow velocity, especially in aortic studies [13]. However, CFD requires patient-specific models and significant computational resources. By integrating image processing algorithms, such as those based on deep learning and finite element simulations, the efficiency and accuracy of hemodynamic predictions can be enhanced [14]. Automatic RAT can be used for laying a foundation for patient-specific radial artery segmentation for CFD analysis. It may facilitate precise, real-time hemodynamic assessments in ultrasound-guided catheterization procedures, ultimately enhancing clinical decision-making.

Despite significant advancements in visualization with ultrasound, several challenges remain to be addressed in RAT. The variations of intra-observer agreements arise from both congenital and acquired patient factors, including anatomical abnormalities, obesity, arterial wall thickening and calcification, luminal stenosis and branches, and subcutaneous emphysema [9]. In addition, the variability in inter-observer agreements, stemming from differences in overall or specific experience levels, has yet to be standardized [15]. Further, RAT is time-consuming and experience-dependent when performing manual annotations and measurements.

The advancement of deep learning has improved individualized treatment by providing end-to-end solutions through accelerated identification within large datasets, thereby reducing disparities in healthcare [16]. It has been previously observed that deep learning is widely applied to the segmentation of carotid and coronary arteries in ultrasonography [17,18]. Distinct from established deep learning-based vascular segmentation, RAT presents a significant challenge due to the wide variability in image characteristics. This variability is attributed to the lack of specialized settings, standardized scanning modalities, and significant inter-individual echogenic variation.

The variability is also recognized as one of the most substantial issues in image segmentation [19]. The prominent U-Net model and its variants, enhancing fully convolutional neural networks, have been adapted to address these challenges [20]. The U-Net skip connections combine the features from the deep decoder and the semantics from the shallow encoder subnetworks. Attention U-Net integrates the attention gate with the U-Net model to emphasize semantically meaningful features while suppressing irrelevant regions, thereby effectively tracking targets of diverse sizes and shapes [21]. UNet++ is modified to improve the capture of fine details through enriching the identification of features from the encoder network prior to the fusion with the semantically similar features from the decoder [22]. The residual module and the high-efficiency channel attention module are integrated into the Res-UNet to improve segmentation performance, especially in cases with blurred borders and irregular shapes [23]. To overcome the limitations of long-range dependencies, TransUNet integrates the transformer and U-Net architectures, enhancing the performance of tokenizing features and cross-attention between features and semantics [24]. UNetXt is a convolutional multilayer perceptron-based network that aims to fulfill the requirements of real-time image segmentation by tokenizing features, shifting channels, and fusing them with decoder semantics [25].

Despite the leading performance of U-Net and its variants in organ segmentation, there is a relative paucity of pioneering research evaluating their feasibility in RAT. The present study aims to train and validate the U-Net and its variant models adapted for RAT, based on novel constructed datasets.

## 2. Materials and Methods

### 2.1. Patients Enrollment and Datasets Formulation

Approved by the Ethics Review Committee of the Chinese Academy of Medical Sciences & Peking Union Medical College Hospital (I-23PJ662), patients were enrolled from May 2023 to Jan 2024 according to the inclusion and exclusion criteria depicted in Figure 1. The patient characteristics were withheld during the entire implementation process of the current study. From the location next to the styloid process of the radius to the confluence to the brachial artery, the radial arteries in both forearms of each patient were continuously scanned and acquired using B-mode ultrasound. The original videos were captured at a resolution of 1200 × 868 pixels for each frame. The frame rates of the videos varied, with approximately 120 videos recorded at 28 frames per second (fps), 30 videos at 25 fps, over 10 videos at 10 fps, and 7 videos at 49 fps. The majority of the videos contained 400 to 600 frames. To fit real-world scenarios and enrich data diversity, videos were acquired using three different ultrasound settings: the SonoSite X-Porter (FUJIFILM Sonosite, Inc., Bothell, WA, USA), the Mindray UMT-150 (Mindray Ltd., Shenzhen, China), and the Wisonic Navi S (Wisonic Medical Technology Co., Ltd., Shenzhen, China). Each ultrasound scanner was equipped with linear array probes with frequencies of 4 to 16 MHz.

### 2.2. Data Preprocessing and Augmentation

All acquired videos were saved and transferred to a workstation equipped with Intel^®^ Xeon^®^ Gold 6132 CPUs (2.60 GHz, 2 processors) and an NVIDIA TITAN RTX 24 GB GPU, along with 128 GB of RAM. After the exclusion of videos with suspected ambiguous manual labels (detailed in Figure 1), images were extracted from videos that presented diverse sites and characteristics along the transition of radial arteries. To optimize computation, the extracted images were converted to a standardized geometry of 256 (height) × 256 (width) × 1 (channel).

The initial datasets, consisting of uniformly formatted images, were randomly divided into training, validation, and test sets at an 8:1:1 ratio. The training set was used for model training, the validation set was employed to avoid overfitting by tuning hyperparameters, and the test set was reserved for evaluating the model’s performance.

To enhance model robustness and generalization, data augmentation was applied in the training set, including random rotation within a range of −10° to 10°, random horizontal and vertical flipping, sharpening adjustments with a random factor up to 2, and brightness and contrast variations within 50% to 150% of the original image. In addition, Gaussian blur was applied with a 50% probability and a blur kernel size of 3 to simulate various capture conditions. Gaussian noise, with a mean of 0 and standard deviation of 0.1, was utilized during each training epoch, along with salt-and-pepper noise with a 2% probability. The dynamic data augmentation of the training set was split into cycles according to the pre-designed size, and each cycle was executed in a randomized sequence. Adam optimizer was applied for the optimization, with an initial learning rate of 10^−3^ and a momentum index of 0.9.

### 2.3. RAT Using U-Net and Its Variants

U-Net and its variants were applied for RAT in ultrasonic short-axis views, including U-Net [20], Attention U-Net [21], UNet++ [22], Res-UNet [23], TransUNet [24], and UNeXt [25]. PyTorch (version 2.3.0) was employed as the framework for U-Net and its variants.

U-Net breaks through the limitations of single-label segmentation presented in previous deep learning models. Through the processes of encoding and decoding, U-Net connects spatial features to their corresponding semantics [20].

Attention U-Net combines the advantages of U-Net model and attention architecture. It is capable of focusing on features of interests, suppressing nonsense information, and dynamically adjusting based on context [21].

UNet++ is modified to improve the performance of detailed segmentation under multiscale circumstances. By incorporating confusion intermediate layers and dense skip connection, the decoder layer is improved in connecting with encoder features through shorter paths [22].

As illustrated in Figure 2, Res-UNet makes innovations in integrating squeeze-and-excitation networks, dynamic channel recalibration and application of atrous spatial pyramid pooling. Thereby, the strength of Res-UNet lies in its ability to highlight relevant features based on semantics [23].

The TransUNet integrates transformers into the U-Net framework by replacing traditional convolutional layers in the encoder, effectively capturing long-range dependencies, as illustrated in Figure 3. This adaptation enhances the modeling of spatial relationships in medical images, which is crucial for accurate segmentation. The decoder pathway retains convolutional layers and skip connections to preserve spatial information and optimize gradient flow. Self-attention mechanisms within transformers enable selective feature extraction, further improving segmentation accuracy [24].

UNeXt integrates tokenized multi-layer perceptron modules to optimize number of layers and sampling strides. Compared to U-Net, it contributes to making image segmentation network more lightweight [25].

### 2.4. Experimental Setup and Statistical Analysis

To evaluate the performance of various deep learning models in segmenting RA regions in ultrasound images, the DSC and the JSC were utilized as performance metrics:(1)$$DSC(A,B)=\frac{2\left|A\cap B\right|}{\left|A\right|+\left|B\right|}$$
(2)$$JSC(A,B)=\frac{\left|A\cap B\right|}{\left|A\cup B\right|}$$
where *A* denotes the RA region identified by the deep learning models, and *B* represents the manually annotated reference region. Both DSC and JSC values range from 0 to 1 (or 0% to 100%), with higher values indicating superior segmentation accuracy.

The loss function is utilized during the model training process to measure the discrepancy between the predicted values and the true values. It serves as a guide for the optimization algorithm, aiding the model in adjusting its parameters to minimize prediction errors. By minimizing the value of the loss function, the model progressively enhances its predictive performance, ultimately enabling it to make more accurate predictions. For U-Net and its variants, a combination of binary cross-entropy (BCE) loss (*L*_BCE_)and dice similarity coefficient (DSC) loss (*L*_DSC_) was used for assessment. The loss function is defined as follows, with both *β* and *γ* set to 0.5.
(3)$$Loss=\beta L_{\mathrm{BCE}}+\gamma L_{\mathrm{DSC}}$$

*L*_BCE_ and *L*_DSC_ are defined as follows, where *A* represents the model prediction and *B* denotes the manual label.
(4)$$L_{\mathrm{BCE}}=-B\log(A)-(1-B)\log(1-A)$$
(5)$$L_{DSC}=1-\frac{2\left|A\cap B\right|}{\left|A\right|+\left|B\right|}$$

*L*_BCE_ is conceptually similar to the cross-entropy loss function but is specifically designed for binary classification without the requirements of the sigmoid or SoftMax functions for mapping input values to the [0, 1] interval. This approach provides enhanced numerical stability compared to standard cross-entropy loss. *L*_DSC_ enhances the effectiveness of image segmentation in scenarios characterized by a significant imbalance between positive and negative samples.

The statistical significance of differences was analyzed using the Kruskal–Wallis test, in terms of DSC and Jaccard similarity coefficient (JSC), for RAT in the test set (*n* = 734). A statistical difference with a *p*-value of less than 0.05 was considered significant. The analysis was performed using IBM SPSS version 27.0 (IBM Corp., Armonk, NY, USA).

## 3. Results

The RAT dataset consisted of 7233 images extracted from the 178 videos of 135 patients. The mean age of the patients was 41.6 years, with an equal distribution of females and males. In terms of ASA classification, 22.96% of the patients were classified as ASA I, 41.48% as ASA II, and 35.56% as ASA III or higher. The patients underwent surgeries in various specialties, including urology (5.93%), vascular surgery (11.11%), gynecology (11.85%), neurology (12.59%), otolaryngology (13.33%), orthopedics (15.56%), and general surgery (17.04%).

Examples of original images and the corresponding automatic RAT results for each model are depicted in Figure 4. The performance metrics for U-Net and its variants are depicted in Table 1. Among the architectures used for training and test, UNext required the fewest parameters, Res-UNet exhibited the fastest overall performance, and UNet++ had the lowest inference time per image.

The loss for the training and validation sets of U-Net and its variants are illustrated in Figure 5. All of the U-Net and its variants achieved convergence during the training and validation processes. Among them, UNext achieves through the fastest pathway and with the lowest loss values.

As illustrated in Figure 6, DSC rapidly improved and stabilized by the 50th epoch during training and validation processes for U-Net, U-Net++, Res-UNet, TransUNet, and UNeXt. However, the Attention U-Net architectures exhibited obvious fluctuations.

Table 2 presents the performance of U-Net and its variants on the test set, evaluated using DSC and JSC. The results are ordered from greatest to least in terms of DSC with the highest values highlighted in bold. From the table, Res-UNet achieves the highest DSC and JSC values, indicating its superior segmentation performance compared to other models, including U-Net, Attention U-Net, and U-Net++. Notably, the DSC values of U-Net++ and U-Net are very close, indicating comparable performance between these two models. Interestingly, the performance of Attention U-Net, UNeXt, and TransUNet is slightly lower than that of U-Net, highlighting that these variants do not necessarily outperform the standard U-Net in this specific segmentation task.

Table 3 presents the *p*-values obtained from the Kruskal–Wallis test comparing DSC across different U-Net architectures. A *p*-value less than 0.05 indicates a statistically significant difference between the corresponding network architectures. Significant differences are indicated with *. For U-Net versus Res-UNet, TransUNet, and UNeXt, the *p*-values (5.76 × 10^−3^, 4.77 × 10^−8^, and 2.24 × 10^−5^, respectively) show significant differences in DSC, indicating that the DSC values of Res-UNet, TransUNet, and UNeXt are statistically different from those of the U-Net. For Res-UNet versus TransUNet and UNeXt, the extremely low *p*-values (8.31 × 10^−18^ and 6.53 × 10^−13^) demonstrate significant differences between Res-UNet and these models.

Table 4 presents the *p*-values obtained from the Kruskal–Wallis test comparing JSC across different U-Net architectures. A *p*-value less than 0.05 indicates a statistically significant difference between the corresponding network architectures. Significant differences are indicated with an *. For U-Net versus TransUNet and UNeXt, the *p*-values (4.49 × 10^−7^ and 1.64 × 10^−6^) reveal significant differences, highlighting that TransUNet and UNeXt differ significantly from U-Net. For Res-UNet versus TransUNet and UNeXt, the *p*-values (2.03 × 10^−16^ and 6.61 × 10^−15^) indicate highly significant differences, emphasizing that Res-UNet has significantly higher JSC than TransUNet and UNeXt.

## 4. Discussion

The development of deep learning improves the elimination of healthcare disparities and unveils novel horizons for precision medicine. Given the outstanding performance in image segmentation of U-Net and its variants, coupled with the requirement to address the heterogeneity of RAT, the current study was conducted. As far as we know, this novel evidence fills the gap in automatic RAT using deep learning architectures.

The results validate the hypothesis that RAT in short-axis views of ultrasonography can be automatically executed by U-Net and its variants. The convergence of *L*_BCE_ and *L*_DSC_ in the training and validation sets indicates the absence of overfitting and underfitting, except TranUNet. Consistent with the previous studies, the compelling results for model weighting were confirmed through the application of *L*_BCE_ and *L*_DSC_, which, respectively optimized the loss functions for classification and quantification [19].

The feasibility of U-Net and its variants for automatic RAT is further supported by the findings of DSC and JSC in test set. In the present study, DSC ranges from 93.14% to 91.08%, comparable to carotid artery segmentation with DSC over 90% [26], and is superior to retinal vessel segmentation with DSC around 85% [27]. The JSC of U-Net and its variants in RAT, ranging from 87.93% to 84.88%, though marginally lower than the performance for common carotid artery segmentation (90.89–84.97%) [28], was significantly superior to breast mass segmentation (63.65%) [29].

Additionally, whereas U-Net and its variants perform competently in RAT, the results indicate that Res-UNet is particularly notable. These results corroborate that Res-UNet improves the accuracy of image segmentation in fetal ultrasonography compared to U-Net and Attention U-Net [30]. A possible explanation could be that the insertion of residual blocks between the encoder and decoder pathways enables a firmer connection between the extraction of interest features and the semantic context.

Although Res-UNet demonstrates the best segmentation performance in this study, the trade-off between its model complexity (e.g., number of parameters and inference time) and segmentation accuracy needs to be discussed. As shown in Table 1, Res-UNet has 24,449,857 network parameters, approximately three times that of U-Net, but its single-frame inference time (0.410 s) is only 0.023 s longer than that of U-Net. Compared to Attention U-Net and TransUNet, Res-UNet has fewer parameters and a shorter inference time. This suggests that although Res-UNet increases model complexity, it still offers acceptable efficiency. However, to achieve real-time RAT in clinical settings, future studies should focus on optimizing the model architecture to further reduce the parameter size. For instance, model compression techniques or lightweight convolutional modules could be explored to reduce computational complexity while maintaining or even improving segmentation performance. Additionally, one may investigate other architectural improvements, such as incorporating depth-wise separable convolutions or efficient attention mechanisms, to further reduce inference time.

Further, this study was conducted based on independent datasets comprising over 7000 images from 135 patients. These datasets represent some of the first large collections of radial artery ultrasound images, curated for the training, validation, and testing of deep learning architectures in image segmentation. The volume surpasses other vascular tracking datasets, which consist of a few hundred images from dozens of patients [31,32,33].

Hence, this pilot study has some limitations and leaves a few questions unanswered. Challenges of the automatic segmentation during RAT exist in certain types of cases, as illustrated in Figure 7. There were challenges in distinguishing between hypoechoic muscle encapsulated with fascia and radial artery presents challenges when applying architectures such as Attention U-Net, TransUNet, and UNet++, as shown in Figure 7a. When multiple vessels or vascular branches were present concomitantly (Figure 7b), the U-Net, U-Net++, and TransUNet models failed to identify the vessel location, Attention U-Net and UNeXt exhibited under-segmentation. ResUNet could detect the vessel, but there was another small over-segmentation region (red arrow in Figure 7b). Despite advancements of the U-Net series in identifying ambiguous boundaries, the enhanced capability to accurately delineate tiny vessels confounded by hyperechoic tissues required further exploration, as shown in Figure 7c. In future work, these challenging cases may be addressed as follows. For the challenge in Figure 7a, multi-modality ultrasound image fusion may be applied, i.e., incorporating multi-modality ultrasound data, such as ultrasound elastography [34] or color Doppler imaging, to complement B-mode ultrasound. As the hypoechoic tissue may have different elasticity or blood flow properties from the radial artery, the differentiation between hypoechoic tissues and the radial artery can be enhanced with elastography or color Doppler images. For the challenge in Figure 7b, the presence of multiple vessels or vascular branches, a rare case in the current model training, results in poor generalization when segmenting such instances. Increasing the number of such cases in the training set may improve the model’s performance in future work. For the challenge in Figure 7c, super-resolution reconstruction techniques may be employed. For instance, generative adversarial networks or super-resolution networks may be utilized to improve the resolution of ultrasound images [35]. This would improve image resolution, making the boundaries of small radial arteries more distinct and thereby enhancing segmentation accuracy.

## 5. Conclusions

This study indicates that automatic radial artery catheterization can be achieved using the U-Net model and its variants. Among these, Res-UNet demonstrates a particularly prominent performance. The potential applications of these findings may help eliminate healthcare disparities, facilitate precision medicine and improve automated procedures in the widely required procedure of radial artery catheterization. The proposed deep learning based radial artery segmentation methods may be used for automatic RAT in ultrasound images. Future research should focus on the remaining challenges, including distinguishing hypoechoic tissues, accurately identifying multiple vessels, and addressing blurred boundaries.

## Figures and Tables

**Figure 1 diagnostics-14-02358-f001:**
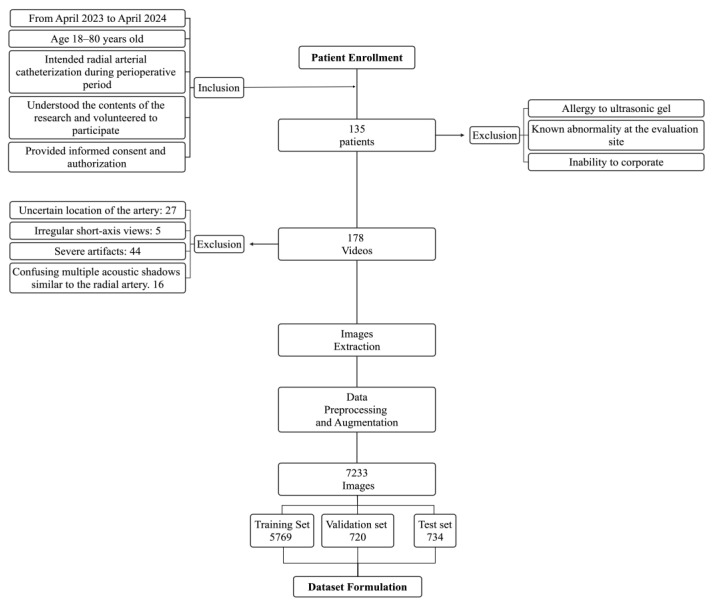
The process of patient enrollment and dataset formulation.

**Figure 2 diagnostics-14-02358-f002:**
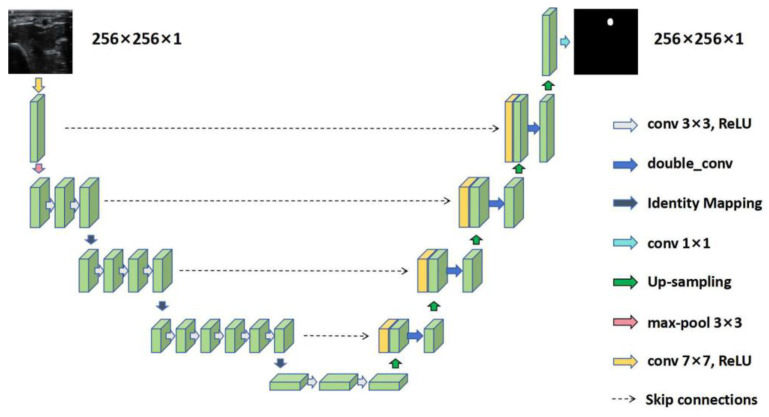
The Res-UNet network architecture for RA segmentation in US images. RA: radial artery; US: ultrasound; conv: convolution; ReLU: rectified linear unit.

**Figure 3 diagnostics-14-02358-f003:**
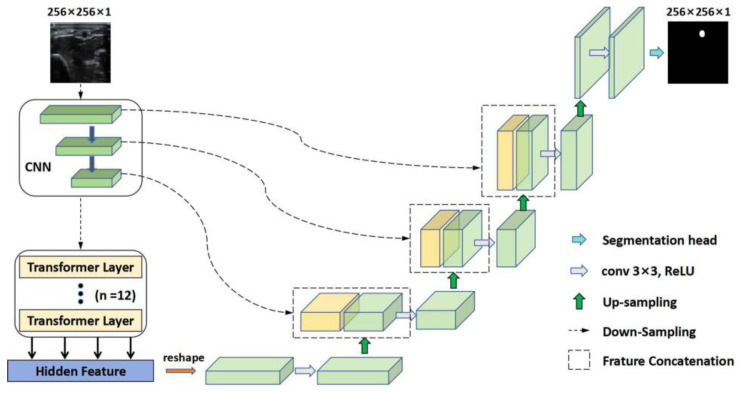
The TransUNet network architecture for RA segmentation in US images. RA: radial artery; US: ultrasound.

**Figure 4 diagnostics-14-02358-f004:**
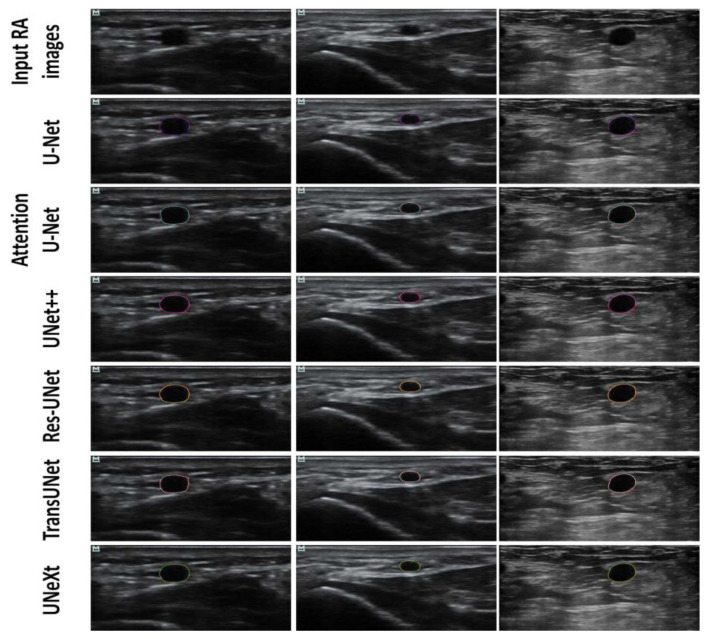
Representative input images (Row 1) and corresponding RAT results performed by U-Net (Row 2), Attention U-Net (Row 3), U-Net++ (Row 4), Res-UNet (Row 5), TransUNet (Row 6), and UNeXt (Row 7). Red contours were labeled by the expert. Contours in colors other than red were the segmentation of the corresponding deep learning architecture.

**Figure 5 diagnostics-14-02358-f005:**
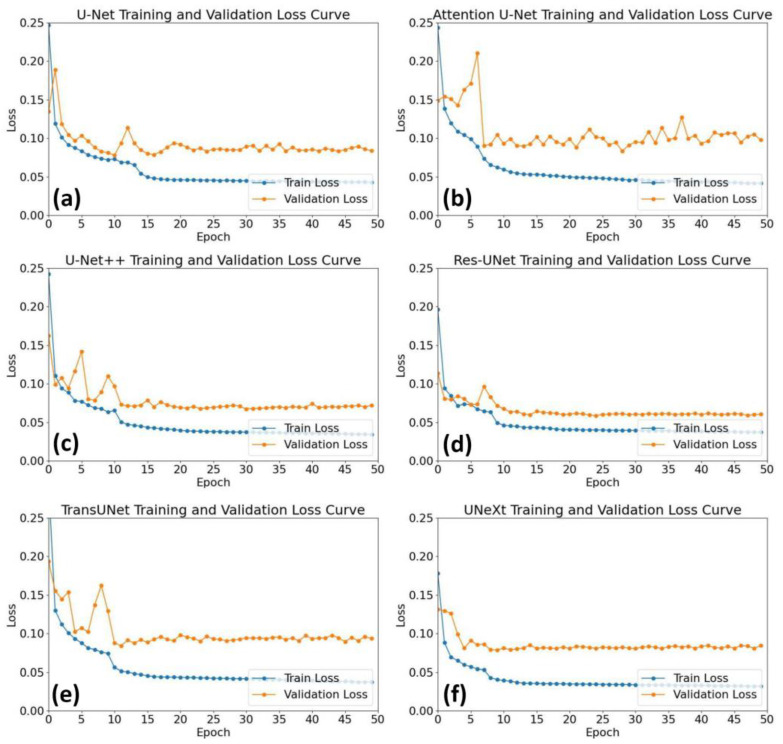
The loss functions of training and validating processes for U-Net (**a**), Attention U-Net (**b**), U-Net++ (**c**), Res-UNet (**d**), TransUNet (**e**), and UNeXt (**f**).

**Figure 6 diagnostics-14-02358-f006:**
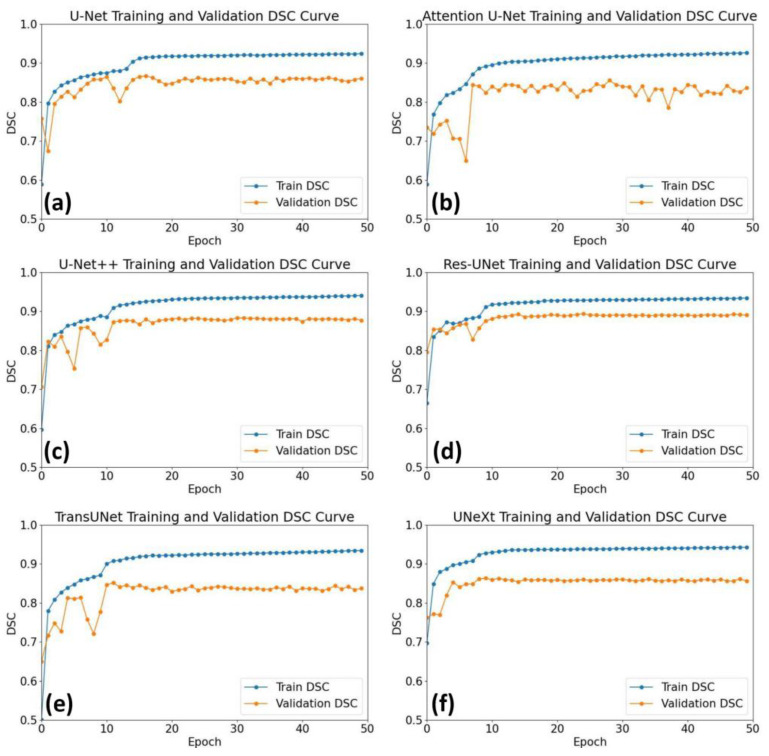
The DSC of training and validating processes for U-Net (**a**), Attention U-Net (**b**), U-Net++ (**c**), Res-UNet (**d**), TransUNet (**e**), and UNeXt (**f**).

**Figure 7 diagnostics-14-02358-f007:**
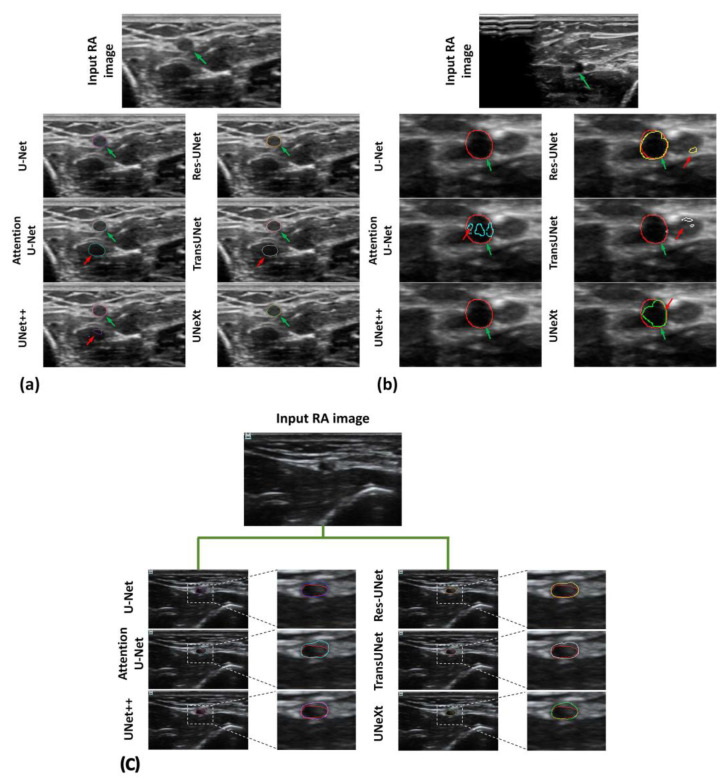
Challenging cases of RAT segmented by the U-Net series. (**a**) Challenges in distinguishing hypoechoic tissues from the radial artery. (**b**) Challenges in accurately identifying the radial artery when multiple vessels or vascular branches are present. (**c**) Challenges in the precise delineation of ambiguous boundaries of the tiny radial artery. Green arrow, accurate segmentation. Red arrow, errors in delineation. Red line, labeled by the human expert. Lines in colors other than red, segmentation of the corresponding architecture.

**Table 1 diagnostics-14-02358-t001:** Performance Metrics in Training and Test Sets for RAT.

	Number of Parameters	Training Time	Inference Time for Each Image
U-Net	7,851,969	19,715.48 s	0.387 s
Attention U-Net	34,877,421	25,938.19 s	0.459 s
UNet++	9,162,753	24,757.11 s	**0.376 s**
Res-UNet	24,449,857	**13,242.36 s**	0.410 s
TransUNet	104,097,149	24,313.91 s	1.868 s
UNeXt	**1,471,633**	17,396.22 s	0.565 s

The optimal value for each metric is highlighted in **bold**.

**Table 2 diagnostics-14-02358-t002:** The DSC and JSC of U-Net and its variants in test set. The highest DSC and JSC are indicated in bold.

	DSC (%)	JSC (%)
Res-UNet	**93.14 ± 3.58**	**87.93 ± 5.57**
U-Net++	91.98 ± 6.31	86.08 ± 8.88
U-Net	91.79 ± 8.51	86.19 ± 10.02
UNeXt	91.33 ± 5.16	84.94 ± 7.93
Attention U-Net	91.20 ± 7.49	85.02 ± 10.17
TransUNet	91.08 ± 7.02	84.88 ± 9.48

**Table 3 diagnostics-14-02358-t003:** Kruskal–Wallis test for evaluating DSC between network architectures. The * indicates statistically significant difference.

	U-Net	Attention U-Net	U-Net++	Res-UNet	TransUNet	UNeXt
U-Net	1	1.57 × 10^−1^	5.36 × 10^−1^	5.76 × 10^−3^ *	4.77 × 10^−8^ *	2.24 × 10^−5^ *
Attention U-Net	1.57 × 10^−1^	1	5.33 × 10^−2^	3.76 × 10^−5^ *	5.20 × 10^−4^ *	1.23 × 10^−2^ *
U-Net++	5.36 × 10^−1^	5.33 × 10^−2^	1	4.32 × 10^−2^ *	4.91 × 10^−10^ *	2.08 × 10^−6^ *
Res-UNet	5.76 × 10^−3^ *	3.76 × 10^−5^ *	4.32 × 10^−2^ *	1	8.31 × 10^−18^ *	6.53 × 10^−13^ *
TransUNet	4.77 × 10^−8^ *	5.20 × 10^−4^ *	4.91 × 10^−10^ *	8.31 × 10^−18^ *	1	2.90 × 10^−1^
UNeXt	2.24 × 10^−5^ *	1.23 × 10^−2^ *	2.08 × 10^−6^ *	6.53 × 10^−13^ *	2.90 × 10^−1^	1

**Table 4 diagnostics-14-02358-t004:** Kruskal–Wallis test for evaluating JSC between network architectures. The * indicates statistically significant difference.

	U-Net	Attention U-Net	U-Net++	Res-UNet	TransUNet	UNeXt
U-Net	1	6.39 × 10^−2^	8.35 × 10^−1^	6.74 × 10^−3^	4.49 × 10^−7^ *	1.64 × 10^−6^ *
Attention U-Net	6.39 × 10^−2^	1	1.07 × 10^−1^	7.34 × 10^−6^ *	4.33 × 10^−3^ *	6.18 × 10^−3^ *
U-Net++	8.35 × 10^−1^	1.07 × 10^−1^	1	3.63 × 10^−3^ *	1.31 × 10^−7^ *	4.49 × 10^−6^ *
Res-UNet	6.74 × 10^−3^ *	7.34 × 10^−6^ *	3.63 × 10^−3^ *	1	2.03 × 10^−16^ *	6.61 × 10^−15^ *
TransUNet	4.49 × 10^−7^ *	4.33 × 10^−3^	1.31 × 10^−7^ *	2.03 × 10^−16^ *	1	7.41 × 10^−1^
UNeXt	1.64 × 10^−6^ *	6.18 × 10^−3^	4.49 × 10^−6^ *	6.61 × 10^−15^ *	7.41 × 10^−1^	1

## Data Availability

Data can be accessed from the corresponding author upon reasonable request.

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
