# Peer review of "U-Net and Its Variants Based Automatic Tracking of Radial Artery in Ultrasonic Short-Axis Views: A Pilot Study"

_diagnostics, 2024, doi:10.3390/diagnostics14212358_

Round 1
Reviewer 1 Report
Comments and Suggestions for Authors
This study evaluates the performance of U-Net and its variants (including Res-UNet, Attention U-Net, UNet++, TransUNet and UNeXt) for automatic radial artery tracking (RAT) in ultrasound images. The models were trained using a dataset of 7,233 images from 178 videos of 135 patients and validated for their accuracy in the RAT. The study clearly shows that Res-UNet achieves the highest performance in terms of Dice Similarity Coefficient (DSC) and Jaccard Similarity Coefficient (JSC) and outperforms the other models in terms of segmentation accuracy.
1-The study rigorously evaluates several U-Net variants and reaches satisfactory conclusions about their performance for medical image segmentation, especially for radial artery tracking.
2- The use of a large dataset (7,233 images from 135 patients) increases the reliability of the results and is satisfactory in the sense that they can be better generalized to real-world clinical applications.
3- The study provides a comprehensive analysis of the model performance using key metrics such as DSC, JSC and loss function convergence, which are essential for evaluating segmentation quality in medical imaging. However
1-The results and motivation of the study should be stated in one sentence in the abstract.
2- Since the study is a simulation and not a real-world application, if the results need to be compared critically, it should be written why real data is not included.
3- Although the study mentions some challenging situations where the models struggle, I do not offer solutions for these situations.
4- Although Res-UNet performs best, the study should further investigate the trade-offs between model complexity (e.g. number of parameters and inference time) and segmentation accuracy.
5- Table 2-4 and Figure 6 should be explained in a detailed and understandable way.
Reviewer 2 Report
Comments and Suggestions for Authors
This is quite interesting. However, a number of issues require clarification before it will be suitable for publication.
Abstract:
This part should be improved. There is almost no description of methodology as well as obtained results. Moreover, abstract is lacking the aim of the study.
Introduction:
Authors should consider adding a paragraph dedicated to image processing and its application in engineering tools for hemodynamic prediction, see e.g.:
10.1016/j.ijcard.2024.132494
https://doi.org/10.1016/j.ijft.2022.100141
Methodology:
1. Authors did not included one of the crucial information. There is no patients description. It should be included in the manuscript.
2. Also, there is no information about the acquisition process. It should be described in details, e.g. data resolution.
Conclusions and Future Scope:
The conclusions – the authors should stress the practical aspect of their work.
Round 2
Reviewer 2 Report
Comments and Suggestions for Authors
Thank you for addressig my comments.
Regarding Q3, the authors focus mainly on deep learning techniques. However, for monitoring of aorta's hemodynamics there exist well established other tools. This includes computational fluid dynamics (CFD), which is one of most common and well verified approaches, often used for this purpose. In my opinion, this method should be briefly described in the paper. I recommend considerig these (or similar) publications.
